# Research on the Application of Mobile Robot in Timber Structure Architecture

Lu Wang [1,2], Tao Zhang [1,*], Hiroatsu Fukuda [2] and Yi Leng [2]

1   Innovation Institute for Sustainable Maritime Architecture Research and Technology, Qingdao University of Technology, Qingdao 266033, China; a9dbb404@eng.kitakyu-u.ac.jp
2   Faculty of Environmental Engineering, The University of Kitakyushu, Kitakyushu 808-0135, Japan; fukuda@kitakyu-u.ac.jp (H.F.); a9dbb418@eng.kitakyu-u.ac.jp (Y.L.)
*   Correspondence: zhangtao841120@163.com; Tel.: +86-159-5489-8078

**Abstract:** The involvement of robots in building construction is already a global trend. Compared with the current stage of construction in which a large number of people are involved, the stability of the robot construction process will greatly affect the construction efficiency and construction accuracy, thus (1) reducing the impact on the environment, hence saving natural resources with other obvious advantages of natural environmental benefits, and (2) reducing construction costs, therefore reducing the economic and environmental benefits of artificial use. This paper proposes a wooden building construction method using a mobile robot, explores the assembly of continuous building components that exceed the robot's static workspace, and completes a simulated construction experiment of a wooden building using this construction method. The experiment was used as a basis to address (1) innovations in the way a wooden building is erected that satisfy the construction logic of the mobile robot, and (2) the ability of the mobile robot to accurately assemble building components in space, including the ability to align them with existing components on site. Ultimately, the completion of this experiment and its construction evaluation demonstrated the superiority of mobile robot construction over manual construction in terms of reduced manual use and increased construction efficiency.

**Keywords:** robotic and automated construction; mobile robot; parametrization design; sustainability

## 1. Introduction

### 1.1. Background of Robotics in Architecture

Advanced digitalization and automation technologies are profoundly impacting almost all manufacturing and industrial sectors and are especially the biggest beneficiaries for those industries that rely on manufacturing processes where parts can be moved around the manufacturing plant [1–3]. Additionally, as the mindset is shifting towards sustainability, reusability, and low-carbon society, robots are becoming increasingly popular in the manufacturing and industrial sectors [4]. According to Tsai and Chang (2012), permanently sustainable development is a term that is now appearing in various areas of social and economic life. Dupuisani et al. [5,6] argued that the sustainability of the industry can be improved by increasing productivity and economic efficiency. Additionally, of all sectors, the construction industry has been an important part of the world's economy and environment and is receiving increasing attention under the global sustainability development [7]. This is not only because buildings account for more than 30% of global greenhouse gas (GHG) emissions and more than 40% of global energy consumption (United Nations Environment Programme 2009) [8], but also because automation in the construction field is considered to be effective in improving construction efficiency and construction stability, reducing worker injuries, reducing waste of resources and manpower, and creating economic value. In addition to the above aspects, robots can also perform construction duties that are impossible or unsafe for humans to perform [9]. Therefore, automation and robotics have

been regarded as leading areas of innovation in construction [8]; the field of building construction, however, has little to gain from them and faces many challenges [10].

In particular, the level of automation is lower in the direction of on-site prefabrication on construction sites compared to the area of off-site prefabrication. In the area of off-site fabrication, digital construction has become relatively common, as exemplified by the experiments on large shell timber buildings conducted by Tongji University in China [11]. Smaller components of a building are made in a dedicated factory and then transported to the building site for final assembly led by manual labor, but it is difficult to adapt these parts to inaccuracies in the building site and the construction process during installation, which breaks the digital process chain between design and manufacturing [12]. Directly on building sites, however, the level of automation is still comparably low. These methods are limited by the fact that the size of the construction machine constrains the size of the building that it can build, and that current construction methods and environments cannot be easily adapted to accommodate these processes. Based on this, recent research has moved in the direction of on-site digital fabrication, where buildings (or building components) are fabricated autonomously on site, generally referred to as in situ fabrication [12].

In situ fabrication is the goal of digitally automated construction in the building industry. A building differs from a structure in that the final assembly of a building has to happen in situ—directly at its final and definite location. In this paper, we emphasize that the experimental object is a complete building, which makes it more necessary for the building to be built on the site, rather than being manufactured in a factory like a structure or building component. To break the range limitation of stationary robot construction, and based on the volume of the complete building, instead, mobile robot construction was chosen, and a multidisciplinary construction platform including digital building information, mobile robot construction, and LiDAR positioning was built. Then, attempts were made to explore a strategy for developing the assembly for the in situ fabrication of mobile robots for wooden buildings.

### 1.2. Advantages of Robotic Construction

### 1.2.1. Energy Environment Improvement

In the context of exploding populations, scarce resources, and global warming, the building industry needs to move faster to develop cleaner, more efficient, and customizable building systems [13]. Construction is a highly dynamic industry, responsible for 40% of global energy consumption, 38% of global greenhouse gas emissions, 12% of global potable water use, and 40% of solid waste generation in developed countries [8,14], largely due to the frequent and lengthy delays that occur in the construction industry. Longer work hours for crews mean that equipment that requires fossil fuels to power needs to run longer, resulting in more pollution [8].

The application of digital and robotic technologies in building construction is considered to be effective in improving energy and environmental sustainability in terms of (1) reducing impacts on the environment: reducing construction waste and optimizing the energy efficiency of machines; (2) improving resource efficiency: integrating the construction process, increasing accuracy, and thus reducing material waste; and (3) creating a high-quality construction and living environment [15]. The obvious environmental benefits of robotic construction are largely dependent on the fact that its automated construction systems are often able to perform repetitive tasks more accurately and faster than manual labor, increasing construction efficiency. The entire construction project will take less time, which means less time is needed to run highly polluting machinery. As construction robots become more efficient, they will make a greater contribution to reducing emissions [13].

### 1.2.2. Economic Environment Influence

The economic environment also plays a key role in the practical application of construction robotics. A reduction in construction activity due to a deteriorating economic environment can be counterproductive to the development of construction automation

and robotics. On the other hand, automated and robotic construction can help increase construction productivity and reduce construction costs, which can increase long-term economic value and is an important driver for the adoption of robotics [16].

The development of construction automation and robotics can increase the long-term economic potential, most importantly in terms of changing the construction labor market. The use of robots, whether in production plants or construction sites, can greatly reduce the number of construction workers and the contact between workers; especially with the impact of coronavirus disease in 2020, the traditional construction methods in the construction industry have been greatly impacted, and the rough development model is not sustainable. At this moment, robotic construction can better adapt to the needs of building construction sites in the era of the epidemic [8]. At the same time, the highly efficient construction characteristics of robotics mean shorter construction times along with shorter working hours, thus reducing labor costs. In addition to the mentioned aspects, robots may perform construction tasks in situations where human labor is not possible, undesirable, or unsafe. In other words, robots can be used on construction sites to replace workers to complete dangerous and heavy labor work and ensure the safety and comfort of workers. Construction such as bricklaying is time-consuming, repetitive, and labor-intensive, often resulting in back injury to workers, making robots good candidates to perform this labor [17]. Based on this, construction robots should ensure safe interaction with workers [9]. Furthermore, the high productivity and quality of robotic construction can effectively reduce construction errors, save construction supplies, and improve the economy of the whole project [18].

Based on the advantages of digital robotic construction in both natural and economic environments, and the lag in its current development in the construction industry, we believe it is necessary to investigate the development of robotics in building in situ fabrication.

## 2. Literature Review

The main research background of this paper is the digitization process of construction manufacturing and the current status of robotics in construction manufacturing, especially the development history and research focus of in situ robotic construction. The premise of this study is that robotics is clearly superior to traditional construction manufacturing in terms of the energy environment, socioeconomics, and the labor market.

### 2.1. History of Robotic Construction Research

In recent years, the use of on-site automation and robotics in the construction industry has not been widespread, as there are not many suitable automation systems [19]. Research on construction robotics and automation started in the 1980s, and since then, developments in robotics sciences have led to a wide range of robotic platforms with varying degrees of autonomous construction modes [9]. In contrast to mobile robots used on the construction site, earlier automated site construction was carried out by stationary robotic systems, a largely automated building integration system initiated by contracting companies and containing a complete integrated information management system from planning to construction, a concept known as computer integrated construction (CIC) [20,21]. However, this system is large and customized for a specific construction site. The first conceptual work on using mobile robots for on-site building construction dates back to the 1990s [9]. As an alternative to stationary robotic systems, the EU Robot Assembly System for Computer Integrated Construction (ROCCO) [18,21] and the Bricklaying Robot for Use on the Construction Site (BRONCO) [22] were among the earliest semi-mobile robotic projects developed for construction sites. Further, the robot "Dimrob" could be used in a leisurely building construction scenario and required the use of static support legs to make it a movable fixed-base robot [17]. The overall work was slow due to the low accuracy of the hydraulic actuators it operates and the need for external laser tracking system support. In the 1980s and 1990s, these aforementioned attempts to integrate robotics into building

construction sites [23] were either expensive and limited flexibility factory-style robotic systems or heavy and therefore slow robotic systems. These reasons may have contributed to the lack of impact and the difficulty in the widespread industry adoption of early robotic systems [24].

At the turn of the 21st century, advances in digital fabrication and the use of programming machines led to a new paradigm in robotic construction, where design knowledge flows became clear and generated a new convergence of design computation and physical artifact fabrication processes [24,25]. Today, many companies are using robotic automation in on-site construction, but mostly applied to very specific subtasks. For example, the "In situ Fabricator" proposed by ETH is a class of mobile robot specifically designed for on-site digital fabrication, but the subject of this experiment is not a complete full-size building [12]. Because of the unstructured nature of the construction environment, which makes human–robot interaction challenging, human proximity and vulnerability in interaction impose severe limitations on human and robotic activities in shared environments [9,26]. The study in [27] used multiple node robots in collaboration to meet the construction needs of large-scale structures. In our paper, we attempted to build a whole house with a limited number of robots using only one mobile arm robot, for other reasons such as positioning. In [4], the research focused on the collaboration of multiple stationary robots to complete the assembly strategy of timber trusses. Compared to stationary robots, our paper mainly emphasizes the use of mobile robots and positioning methods, which can perform a larger range of operations and are more suitable for complete building construction. In [28,29], quasi-fixed-base full-size mobile systems capable of printing large-scale foam structures for whole-house 3D printing were presented, but the limited movable fixed-base units do not require robotic repositioning. These challenges mean that automated construction using mobile robots is not yet ready for the commercial market [9]. To date, no robotic platform has been able to fully satisfy the requirements for autonomous mobile robotic architectural construction of an equal scale.

### 2.2. Research Focuses on Mobile Robotic Construction

In the field of automated in situ construction, there are two main categories of construction robots: stationary robots and mobile robots. In comparison to the limited workspace of stationary robotic systems, mobile robots can fabricate structures bigger than their static workspace. At the same time, mobile robots meet both the dexterity and agility requirements as well as the high payload requirements of construction tasks, which expand the scale and enrich the complexity of constructible buildings. However, in the context of robotic in situ fabrication, the robotic processes are challenged by a variety of external influences and uncertainties in their immediate environment [17,24]: Firstly, construction requires precise positioning, and in comparison with their digital blueprint, building site environments often exhibit deviations and dimensional tolerances, which can occur when mobile robots have no common frame of reference with the construction and if the digital model of a building site does not capture a very accurate and detailed depiction of the actual as-built conditions. Secondly, construction sites are also highly complex working spaces and are not static but evolve over time and constantly change with the ongoing progress of construction, where displacement and mechanical work requires high dexterity. Moreover, one goal of automated construction is to prevent worker injuries; therefore, construction robots should ensure safe interactions with workers [9].

Therefore, this paper attempted to propose solutions to the problems of site positioning, human–robot interaction, and the design and completion of a mobile robot site construction simulation experiment for a wooden building. Additionally, this study verifies the feasibility and applicability of the wooden building constructed with a mobile robot from the experiment. This challenge raises many different levels of questions regarding design, construction, and research. Finally, the system needs to be integrated into the layout system and architectural design software for seamless interaction between

design and construction. Overall, we find that addressing robotic fabrication itself poses a multidisciplinary challenge.

## 3. Materials and Methods

In this paper, we take the in situ construction simulation experiment of a complete wooden building as an example to explore the possibilities and challenges of continuous building construction that exceeds the static workspace of robots, as shown in Figure 1, whose building structure, erection method, and structural connections need to be adapted to the construction steps of the mobile robot. The on-site construction experiment of the mobile robot has three basic focuses. Firstly, it is a construction process integrated with an automated method, where the parametric building model is adapted to the mobile robot's construction method and the scanned and sensed construction site. Secondly, the experiment also integrates and validates the scanning and positioning system during the experiment so that the robot can perform construction tasks from multiple locations. Finally, it is hoped that the experiment will allow the exploration of assembly strategies for the development of on-site fabrication of wooden structure movable robots.

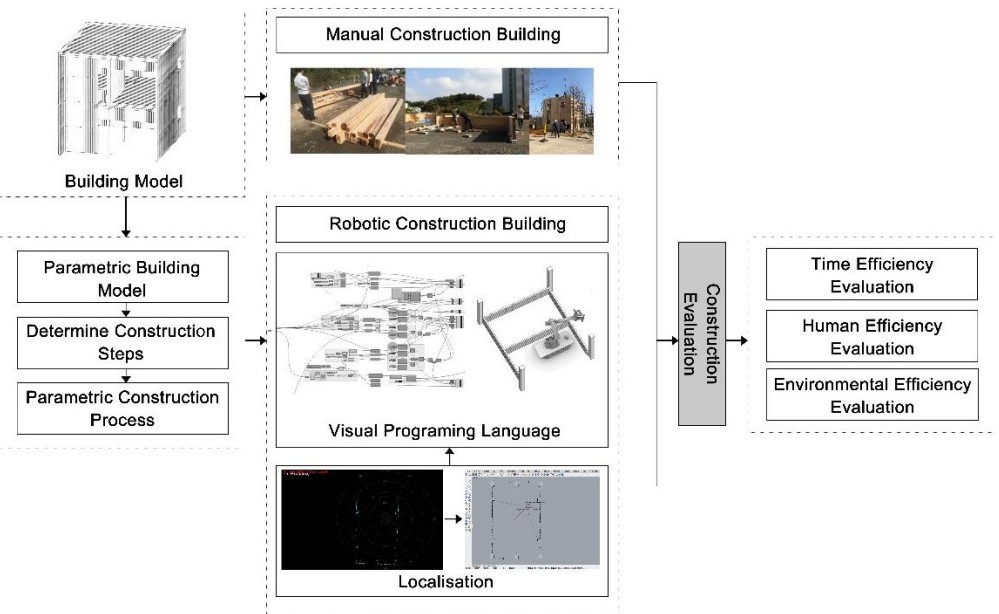

**Figure 1.** A conceptual framework for the robotic construction of a timber house.

The following procedure of this simulation experiment is mainly used to demonstrate the capabilities and advantages of mobile robots for on-site construction. Sections 3.1–3.3 introduce the experimental objectives, the mobile robot and LIDAR properties, and construction material characteristics required for the experiment. Sections 4.1 and 4.2 describe the construction process of the mobile robot and the spatial localization capability of LIDAR in the experiment to validate the functionality of this fully integrated construction system. The construction evaluation in Section 4.3 confirms the effectiveness of robotic construction in reducing labor and increasing efficiency. The challenges of the experiment and future research directions are described in Sections 5 and 6.

### 3.1. Objective

The target building was a two-story wooden building, which was manually constructed. This paper focused on the feasibility and specific construction steps of constructing this wooden building on site with a mobile robot and compared the advantages and disadvantages of manual construction and mobile robot construction in terms of software modeling, construction efficiency, and construction difficulty.

### 3.2. Experimental Robotic Set-Up

The main construction tool for this simulation experiment was a mobile robot equipped with an end effector. Its end effector is capable of performing the construction tasks for each operation step needed for this experiment and consists of a gripper for picking and placing procedures and an air nail gun for performing nail pressing procedures, as shown in Figure 2.

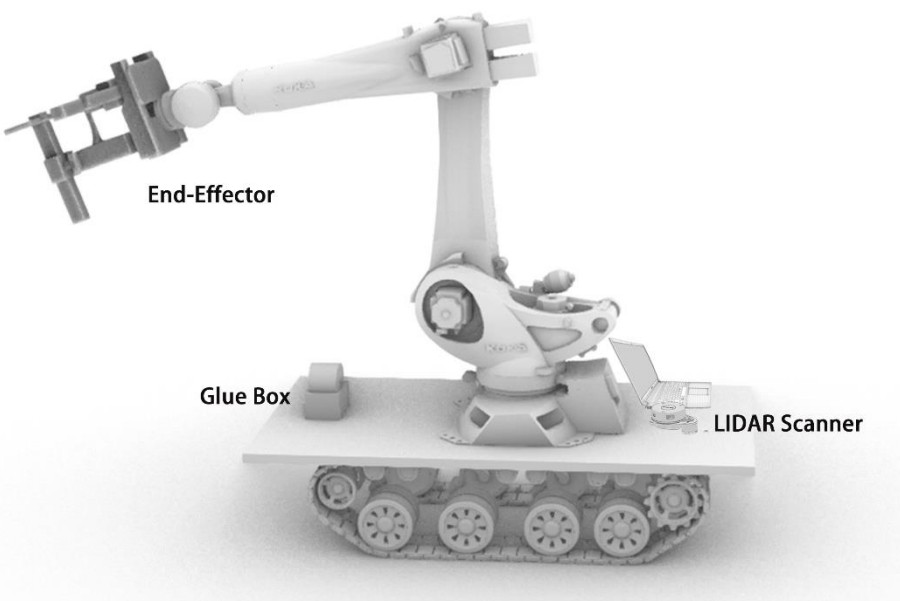

**Figure 2.** The mobile robot is equipped with an end effector.

To accomplish the task of repositioning the robot after it has moved during the construction process, a LIDAR laser range scanner and a laptop computer connected to it need to be placed on the platform of the mobile robot, as shown in Figure 2. Once the radar is operational, it can scan the site environment in real time and can output the point cloud data through the software.

The main software environment that the experiments rely on was implemented in grasshopper, which is a visual algorithm editor plugin for rhinoceros and is commonly used in the architectural design industry [4]. Additionally, it enables software interaction functions for architectural design, robot task execution, and LIDAR ranging.

### 3.3. Building Material System

The target building was constructed of wood, mainly because of the environmental properties and economic benefits of wood, and because the choice of building materials is critical to the construction of renewable buildings. Wood is a major renewable resource for the construction industry and was selected as the preferred building material for the following reasons. Firstly, in terms of the sustainability of wood, a building material available in the local market, Japanese cedar, was selected for this experiment, as it is the most important economic artificial tree species in Japan, and its artificial forest has a short cultivation period for quick maturity and obvious environmental and economic benefits [25]; further, waste wood is a valuable resource that can be recycled. Secondly, it is relatively cheap compared to other construction materials suitable for outdoor applications. Thirdly, depending on the material properties of the wood, it has high durability and strength compared to its weight and can be flexibly processed into different shapes and lengths, making it an adjustable component in construction [11].

## 4. Results

### 4.1. Overview of the Building

#### 4.1.1. Design Objective

The target building was a wooden structure two-story building with a construction area of 49.93 m². To improve construction manufacturability, the floor plan was designed in a simple form for easy construction, as shown in Figure 3. The building was constructed using 100 mm² cross-sectional, 600 mm-long timbers. Having the same length of timbers not only facilitates the processing and cutting of materials in the preliminary preparation but also makes it easy for the discharge operator to add timbers to the robot during the construction process.

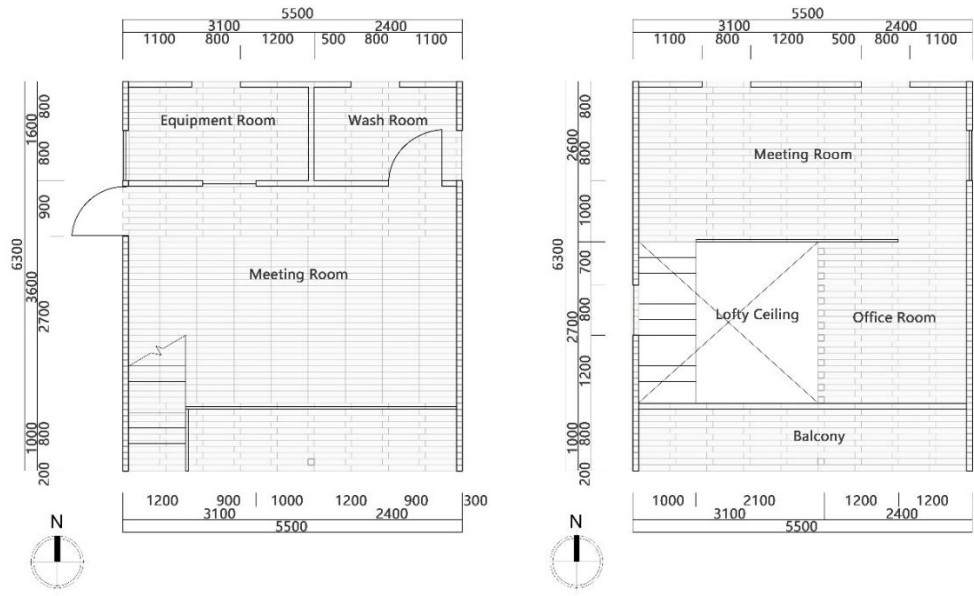

**Figure 3.** The two-story floor plan of the building.

#### 4.1.2. Fabrication Sequence for Robot Fabrication

The building was built in eight units perpendicular to the ground from A to H, as shown in Figure 4. Considering the building range of robots at the same location, each unit was set to 7–9 layers. When building in groups, each unit is built lying flat on the ground, and after the robots finish building, the units are then erected vertically on the ground and connected.

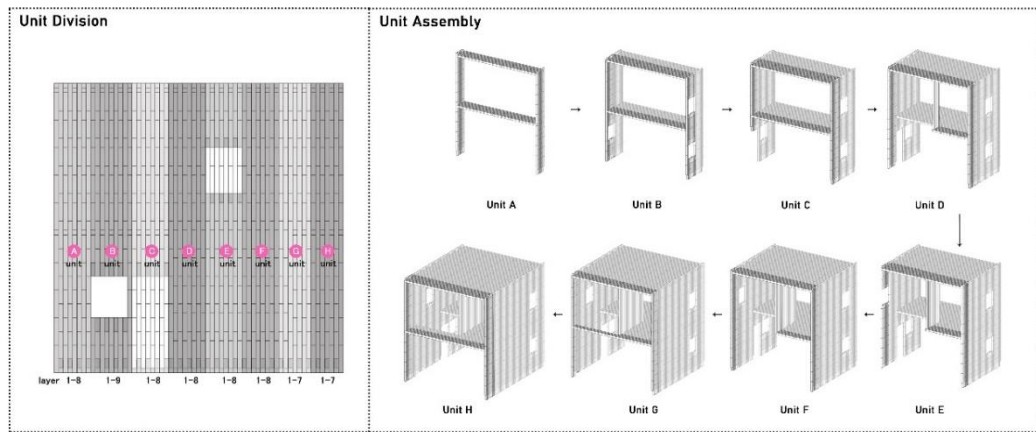

**Figure 4.** Unit division and unit assembly.

The arm length of the construction robot is limited, and in the face of a large building scale, the mobile robot needs to complete the construction of the complete units of the building at different locations. Therefore, a staggered arrangement of timbers in the horizontal direction was used here to form a new structure that is different from a manually constructed structure. In this experiment, a unit needs to be built in four groups, during which the mobile robot needs to undergo three changes of building positions, as shown in Figure 5.

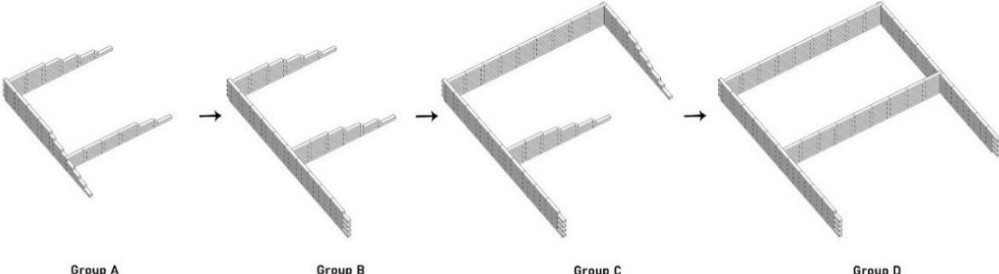

**Figure 5.** Mobile robot construction in four groups for each unit.

### 4.1.3. Structure Construction

The construction process was conducted in sub-units, all operated by mobile robots. The connection between the timbers was achieved with glue and nails, as shown in Figure 6. Therefore, the steps of nailing and glue application are also the main tasks conducted by the robot arm.

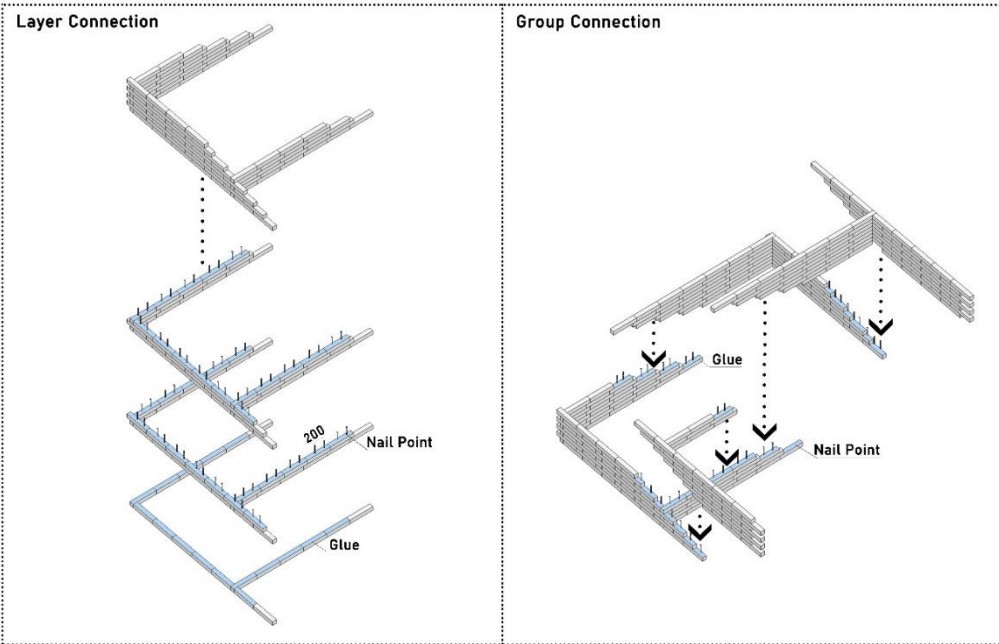

**Figure 6.** Mobile robot construction in four groups for each unit.

### *4.2. Construction Procedure*

### 4.2.1. Determine the Operable Range of the Robot

A total of 2168 timbers were required to construct this building with a length of 6.3 m, a width of 5.5 m, and a height of 6.2 m. The building is divided into units, with the largest number of layers in a unit of nine; the height of the units is 0.9 m. Before the construction process begins, the first step is to determine the operable range of the mobile robot in the same location as the construction range, as shown in Figure 7.

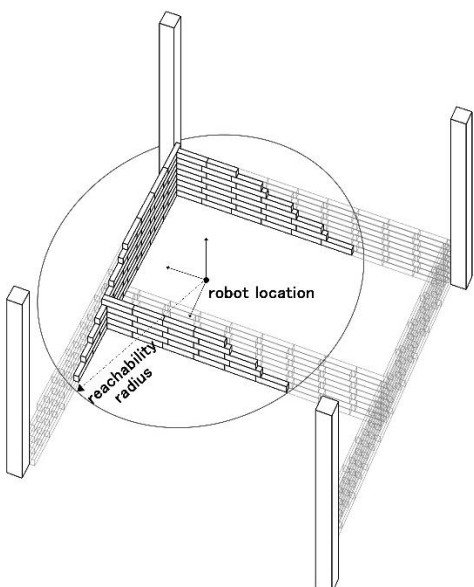

**Figure 7.** The robot location and its reachability constraints.

After this, to satisfy the minimized number of robot relocations, it can be precomputed that the entire assembly sequence needs to be built by dividing each unit into four groups. To complete this assembly sequence, the mobile robot needs to be displaced three times and complete the construction at each of the four computed localizations, as shown in Figure 8, after which the robot can interactively generate the assembly sequence at its current location and within its static range.

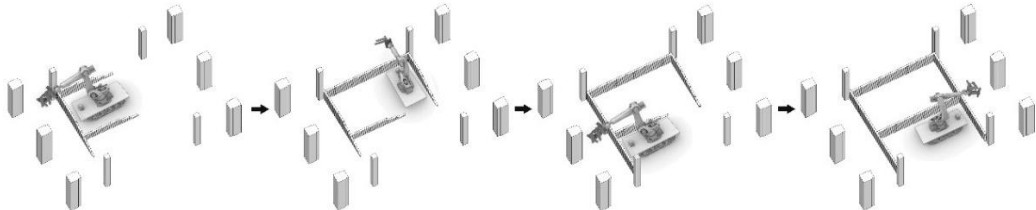

**Figure 8.** Snapshots of one unit with four groups built by the mobile robot.

4.2.2. Robot Construction Steps

The robot completes the assembly sequence with the same steps in each static range to minimize the number of manual steps in the construction process. Here, the assembly sequence conducted by the robot has four main steps: picking timbers, applying glue, placing timbers, and nailing. The four steps are described in detail below.

Pick: Unlike the timbers of varying lengths used in manual construction, this experiment uses timbers of the same size and does not require the operator to deliberately identify their dimensions. The picking of the timbers is the first step in this assembly sequence. The gripping clamps on the robot's end effector are designed to have the same picking point and are at the midpoint of each timber.

Glue: The connection between the timbers requires the application of glue and nails. In the glue application step, the robot grips the timbers and passes over the roller of the glue box in a line with the topmost point of the roller and the length of the timbers as the length. The glue box was designed as a container filled with a water-based polymer wood adhesive and a roller on which the glue can be picked up, as shown in Figure 9.

Nail: In addition to the glue application, the connection between the timbers is enhanced by nailing. The robot end effector is equipped with an air nail gun that can be fitted with nails longer than the length of the timbers. For each timber, the air gun is used to nail two nails at a distance of 200 mm, and the nailing operation is performed by holding

the timber in the set position while the air gun shoots the nails. After this, the gripper releases the timber and the robot rotates 180° on the A6 axis and then performs the same operation, as shown in Figure 10.

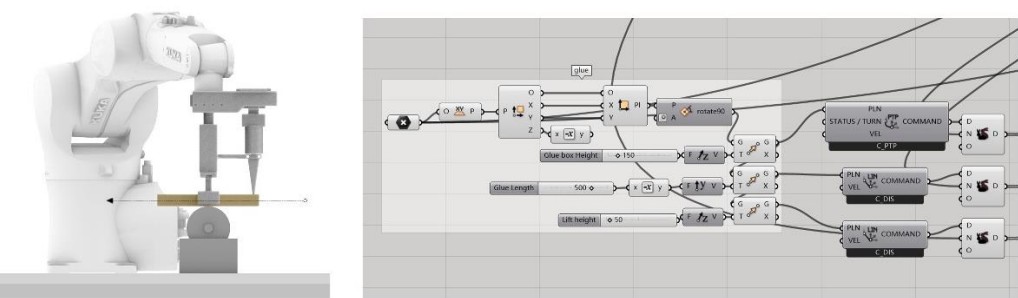

**Figure 9.** Robot construction steps: glue.

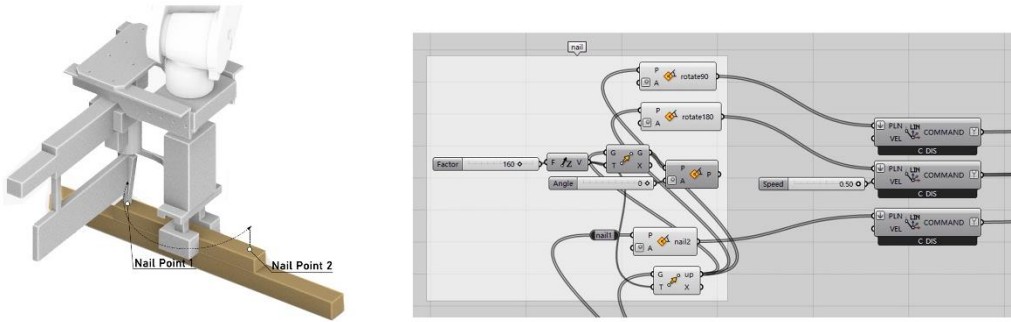

**Figure 10.** Robot construction steps: nail.

Place: In case of a collision with adjacent timbers in the process of placing, instead of the LIN motion method of straight-line motion between two points, the robot needs to run the moving route of clamping the timbers with the robot's A5 axis as the main rotation PTP + LIN motion method, as shown in Figure 11.

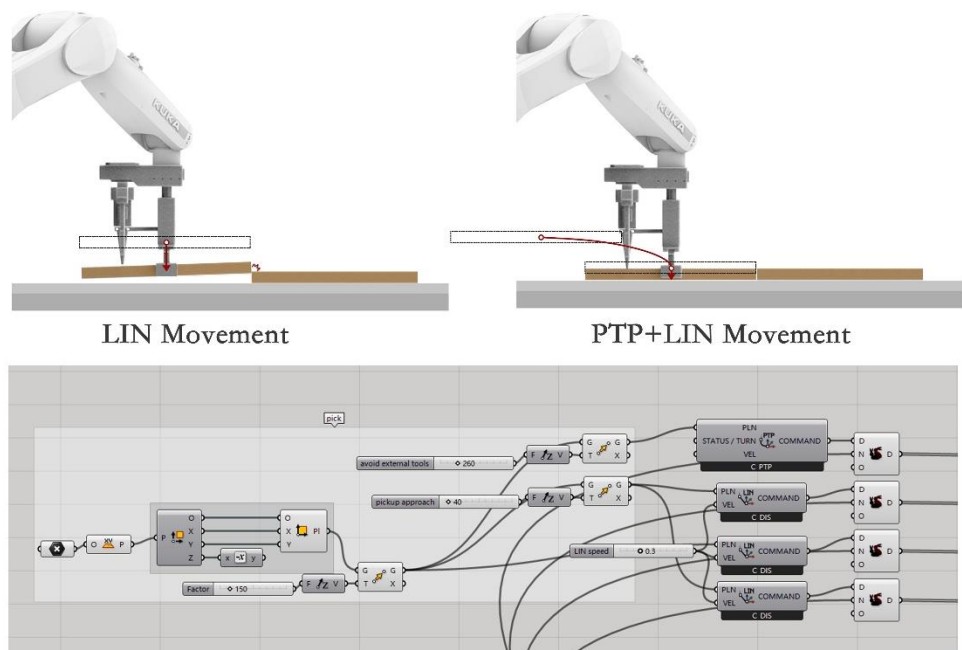

**Figure 11.** Robot construction steps: pick and place.

### 4.2.3. Mapping, Alignment, and Localization

The first step is mapping and alignment. Before starting the fabrication process, the building site needs to be mapped by the robot from a set location. This mapping is achieved by executing a sweeping motion by a LIDAR instrument placed on the mobile robot platform to capture an initial point cloud for each static location. Next, the existing model of the experimental construction site is aligned with the point cloud obtained from the scan.

In this experiment, the scanned building construction site dimensions may deviate from the designed CAD model dimensions, and these deviations will affect whether the designed building component locations match the realistic construction site during building construction, meaning it is necessary to scan and confirm the actual site environment before starting construction. Within this experiment, the key objects of the site are represented by additional columns immediately adjacent to each building unit, and the positions of these key objects in the CAD model are aligned with the actual scanned point cloud positions. When their as-built poses are fed back into the architectural design and planning environment and errors occur, the CAD model of the building site needs to be updated to align the design with the actual environment.

The point cloud of the construction site scanned using LIDAR is shown in Figure 12a, and the columns marked in blue on the right are the key objects of the design. Using the interaction between the LIDAR software and the parametric modeling software, the CAD model of the design and the scanned point cloud can be matched and aligned by key objects in the same software, as shown in Figure 12b.

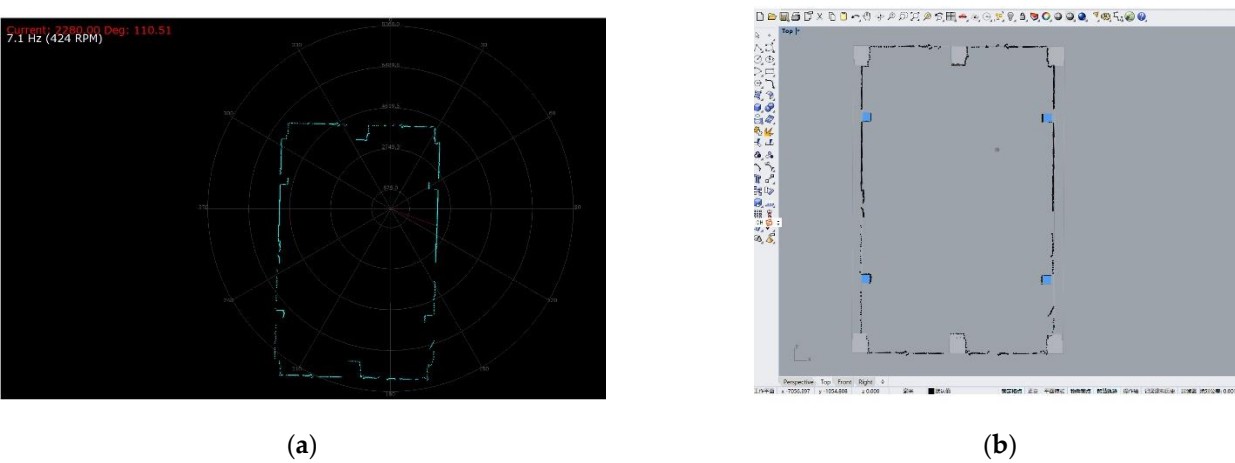

(**a**)                                                             (**b**)

**Figure 12.** Mapping (**a**) and alignment (**b**) of the construction site.

After the site environment is determined to be aligned with the designed CAD model construction site, the robot needs to be relocated for each displacement. This requires two radar scans at each designed positioning point and the use of key objects as identification objects. First, the scanned point cloud of each positioning point is obtained and interacts with the modeling software to generate a construction site map for each point before the experiment starts the construction step; after starting the construction, each displacement of the robot is rescanned after reaching the design position, and the key objects are used to align the two scanned point clouds before and after, as shown in Figure 13. If there is an error between the two scans, when the point cloud of the construction environment is aligned, the deviation of the center point of the two scans is the deviation of the robot's position after displacement. Based on the angle and distance of the difference between the two points, the deviation between the displaced robot position and the original design construction positioning point can be calculated, and the relationship between the position of the construction object and the original robot construction point in the designed CAD

model can be modified accordingly, as shown in Figure 14, so that the experiment can be carried out accurately after the displacement.

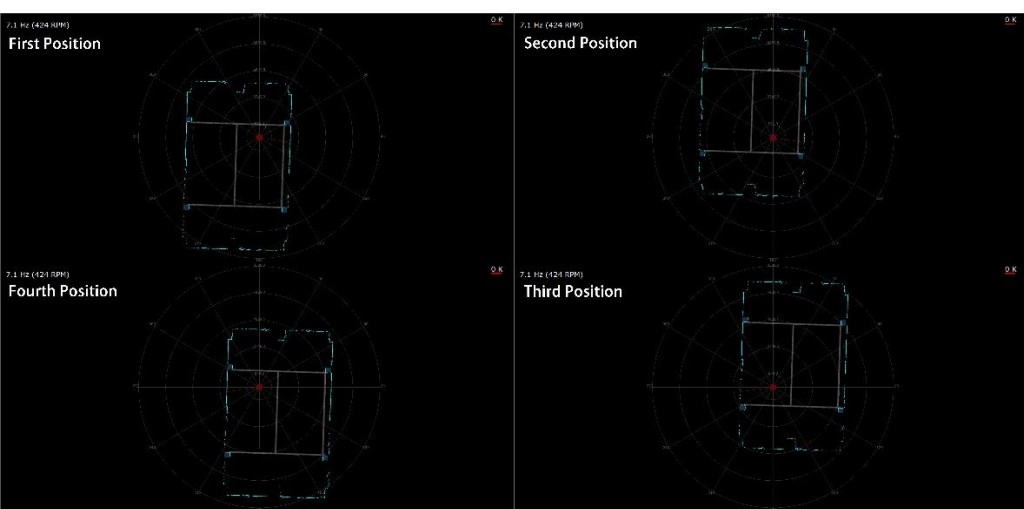

**Figure 13.** Scanning and alignment of the robot's four positioning points.

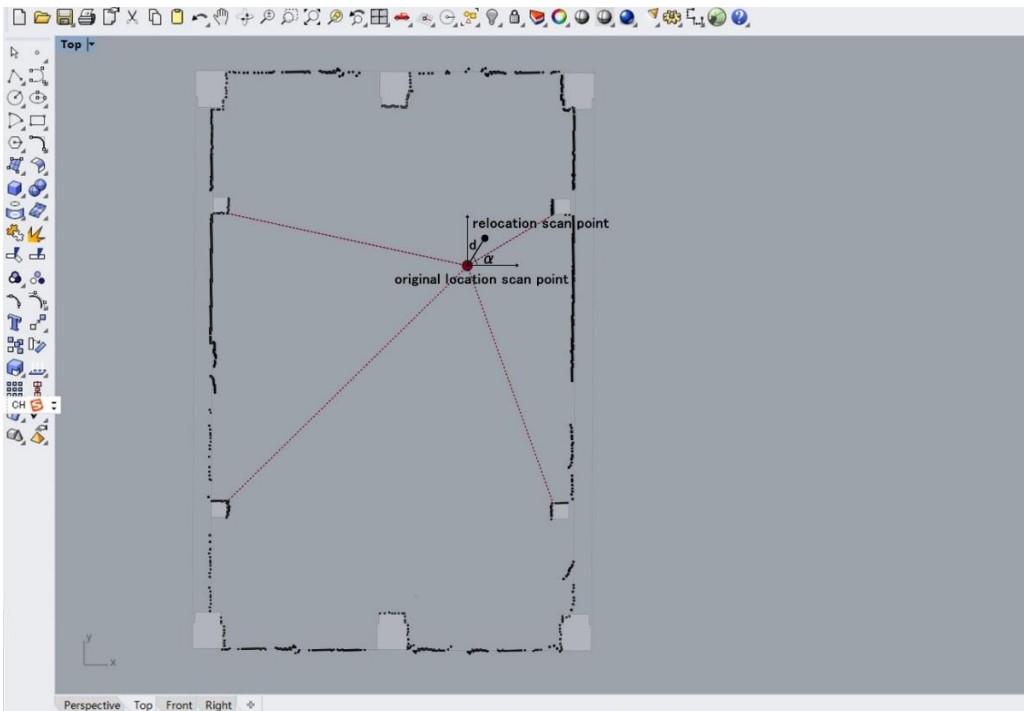

**Figure 14.** Mapping and alignment of the construction site.

After the calibration process of mapping and aligning the construction area and adjusting the position of the construction object to the robot's original point, the next construction is ready to proceed. The mobile robot performs a static operation at each positioning point and then moves to the next positioning point and repeats the calibration process to complete the next set of building construction. In this process, each step of the robotic construction requires the construction instructions to be converted into digital information that the robot and LIDAR instrument can receive. This is a series of digital information that can be interacted with and adjusted in time to ensure that the entire construction process can be carried out smoothly.

The following flowchart outlines the control loop for this fabrication process, which controls the robotic nodes to complete the building fabrication, as shown in Figure 15.

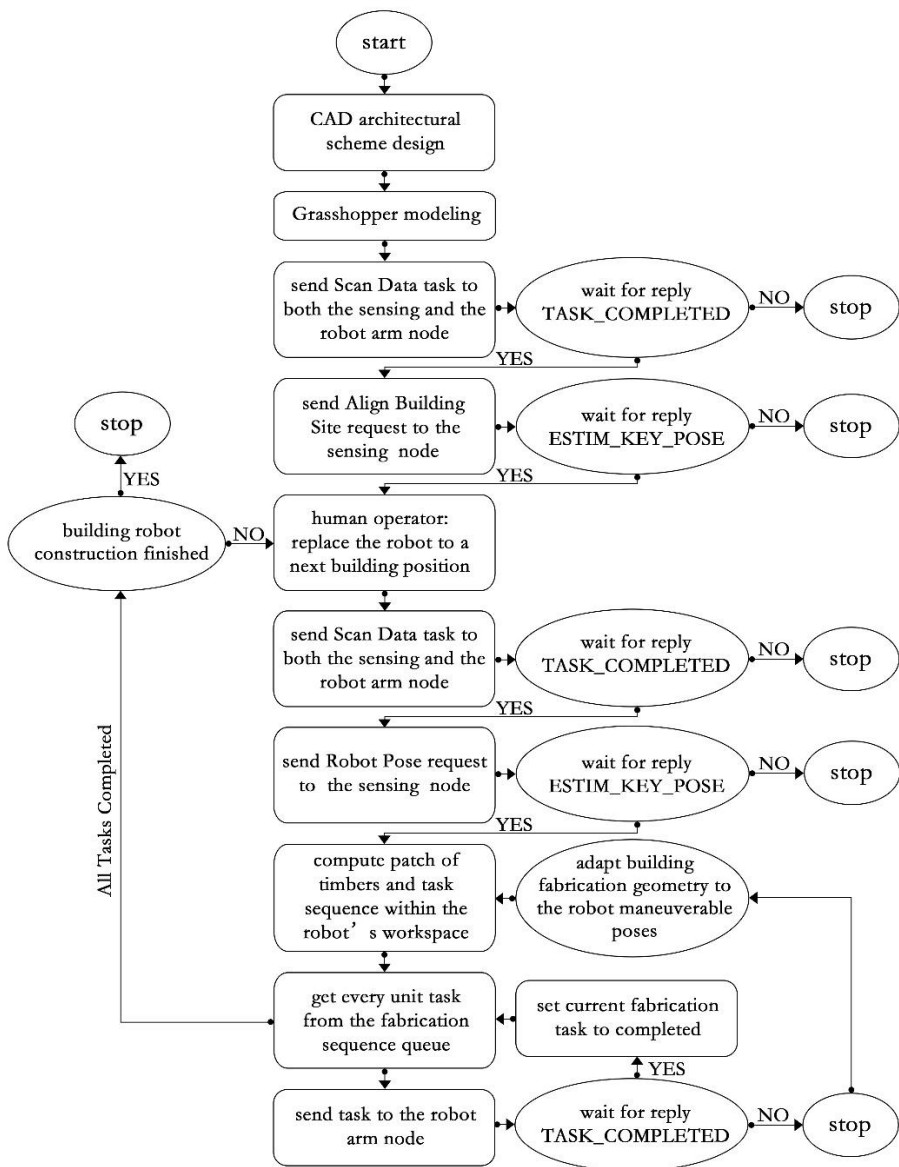

**Figure 15.** Flowchart for the timber building's fabrication control loop.

*4.3. Construction Evaluation*

Construction evaluation is one of the main focuses of this experiment. The comparison between manual construction and robotic on-site construction experiments allows the advantages and disadvantages of two different construction methods for the same timber building to be identified. The following section shows the advantages of robotic construction in terms of the natural and economic environment and the reasons for choosing robotic construction in terms of quantitative data.

The manual construction building and the robotic construction building in this experiment have the same building plan, but due to the different construction methods, the building structure and construction process are different and have a greater impact on the final construction efficiency.

4.3.1. Manual Construction

There are many uncertainties in manual construction, which can be influenced by site conditions, the construction personnel, and other uncertainties. Among these, the construction personnel have the greatest impact on construction efficiency, and their number and efficiency directly determine the project construction time, as shown in Table 1.

**Table 1.** Table of numbers relating to the manual construction of each unit.

| Unit | A | B | C | D | E |
|---|---|---|---|---|---|
| Construction time (min) | 115 | 111 | 137 | 153 | 183 |
| Number of timbers | 72 | 97 | 83 | 70 | 93 |
| Number of layers | 7 | 7 | 8 | 9 | 7 |
| Average per layer | 16.43 | 15.86 | 17.13 | 17.00 | 26.14 |
| Number of people | 13 | 8 | 9 | 8 | 4 |
| Time per layer × people | 213.57 | 126.86 | 154.13 | 136.00 | 104.57 |

This construction was conducted by students lacking knowledge of the construction process and proficiency, and the number of students working on the construction varied greatly. From Figure 16, the total construction time of unit A including manual factors is the longest, and that of unit E is the shortest, so the efficiency of unit A is the lowest and the efficiency of unit E is the highest. It can be concluded that in this construction, each unit does not require much construction personnel, and the minimum number of construction personnel for unit E is 4, but the average efficiency is the highest; meanwhile, the maximum number of participants in unit A is 13, which shows that more than a certain number of construction personnel will affect the construction efficiency of the whole project. Unit A was the first construction unit, and the construction personnel were not familiar with the construction process, so they worked inefficiently and had the lowest average efficiency.

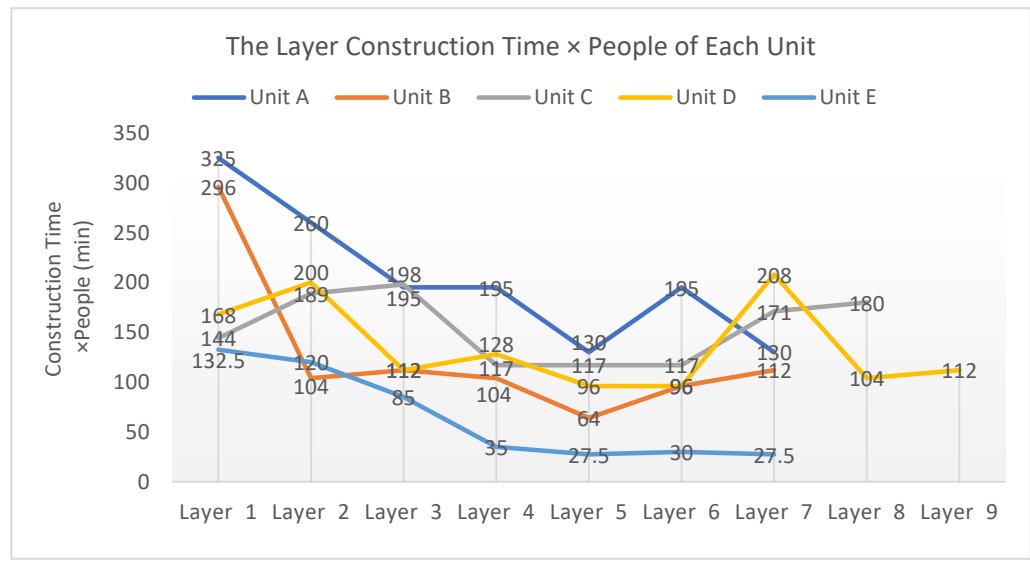

**Figure 16.** The layer creation time × people of each unit by manual construction.

### 4.3.2. Robotic Construction

Like manual construction, robotic construction is conducted one by one for each unit, except that each unit is divided into four groups. The construction time in Table 2 represents the time required for the robot to build the timbers according to the specified steps within the static working range of each group. Each step of the construction process has been digitized and executed by the robot, and the construction time can be calculated from the design program and is relatively stable. The robot moves and relocates three more times during the construction of each unit, and the time for this step is also stable and calculated into the construction time of the whole unit.

**Table 2.** Table of numbers relating to the mobile robot construction of each unit.

| Unit | A | B | C | D | E | F | G | H |
|---|---|---|---|---|---|---|---|---|
| Construction time (min) | 303.10 | 292.70 | 268.54 | 261.09 | 244.62 | 224.20 | 234.74 | 295.58 |
| Robot relocation time (min) | 90 | 90 | 90 | 90 | 90 | 90 | 90 | 90 |
| Number of timbers | 305 | 294 | 276 | 269 | 248 | 231 | 241 | 304 |
| Number of layers | 8 | 9 | 8 | 8 | 8 | 7 | 7 | 8 |
| Average time (min) per layer | 49.14 | 42.52 | 44.82 | 43.89 | 41.83 | 44.89 | 46.39 | 48.20 |
| Number of people | 2 | 2 | 2 | 2 | 2 | 2 | 2 | 2 |
| Time per layer × people | 98.27 | 85.04 | 89.63 | 87.77 | 83.65 | 89.77 | 92.78 | 96.39 |

Throughout the robotic construction process, operators need to be involved in a limited number of steps, so the number of operators required is small and the overall construction time is stable. This also shows that after meeting the number of operators needed for construction, continuing to increase the number of people does not improve the overall construction time. This shows that robotic construction is stable in terms of construction time and labor usage.

To accommodate the construction of a mobile robot, short timbers of the same size are selected for staggered and overlapping construction. Compared to manual construction, the number of timbers increases, the number of robot repetitions of construction increases, and the total construction time of each unit becomes longer. However, the number of operators required for the whole construction process is smaller, and the total labor time is reduced, as shown in Figure 17. Therefore, robotic construction is useful for reducing the use of labor and increasing the overall construction efficiency including the labor factor.

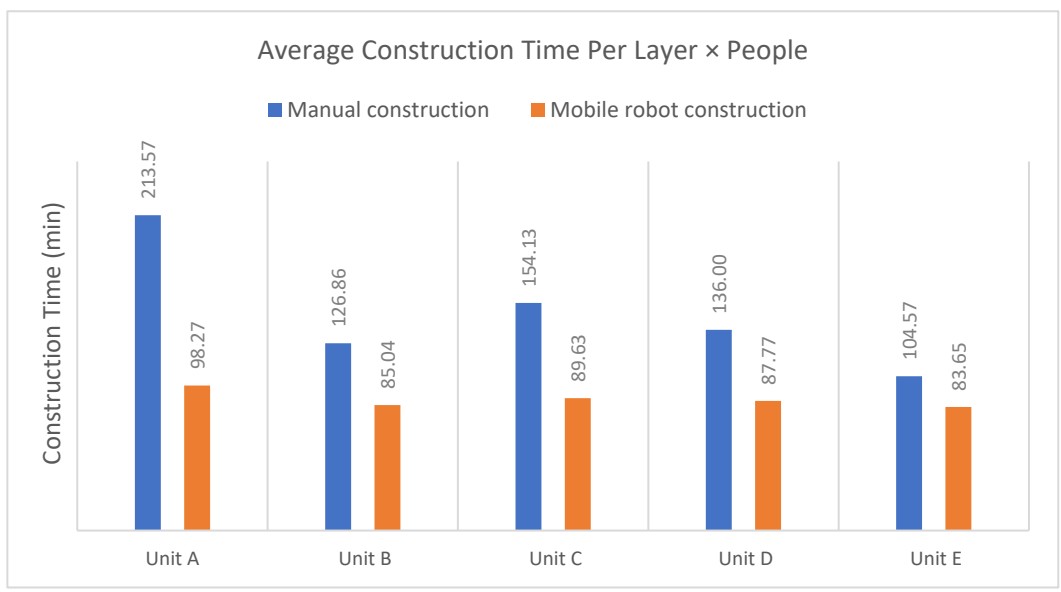

**Figure 17.** Comparison of construction time per layer for the first 5 units × people between manual and robotic construction.

## 5. Discussion

Based on the above experiments, it is clear that an integrated multi-science application approach combining robotics, laser scanning, and positioning technology with building construction makes robotic on-site construction possible and marks the possibility of integrating parametric design and robotic manufacturing processes into building construction projects. The results of this experiment can be summarized as follows: First, the results of

the experiment confirm that robotic construction contributes to the reduction in labor use and material waste, thus improving the energy environment and economic environmental benefits. Second, it developed a mobile building unit construction and assembly strategy that allows the building units to be assembled continuously at different robot construction locations with a minimum number of robot relocations through overlapping, staggered construction. Third, the interaction between the laser scanning software, the robot software, and the modeling software allows the building site to be mapped and the CAD model of the building to be adapted to the true dimensions of the site scan while also allowing the robot to localize itself with sufficient accuracy to complete its building construction tasks. Fourth, the parametric design of the robot's arm and end effector allows it to automate multiple construction steps of building construction. According to the results of the experiment, robotic construction improves the sustainability of the construction industry in terms of increasing production efficiency and economic efficiency [5,6]. By comparing the experiment with manual construction, it can be seen, firstly, that the total construction time is reduced, which is a way to improve the sustainability of construction by increasing construction efficiency, and, secondly, that fewer construction workers are needed, and less timber is wasted, which is a way to improve the sustainability of construction by increasing economic efficiency.

For the future of robotic wood construction on site, the following is also clear from the experiments: First, the experimental simulation of the construction site environment is relatively simple, but the real construction environment will be more complex and need to deal with the uncertainty of the unexpected situation so that the robot can deal with the complex situation of the construction site, which will be the focus of subsequent research. Secondly, for the positioning method using LIDAR, the errors do not accumulate because the obtained point clouds are matched in relation to the reference point clouds obtained from the same initial values; however, this location method requires the surroundings to remain largely unchanged and does not exactly match the changing environment of the construction site [30]. Therefore, an attempt can be made to scan only the building components, instead of scanning the environment. However, its ability to provide the desired accuracy is debatable [24]. Third, the research object of this experiment was a wooden structure building, and the robot operating system and its end effector can complete the nailing and glue application connection method, but different building structures have different construction methods, so it is necessary to consider the adaptability between the robot operation and robot end effector under different construction methods, which means that the exploitation of the end effector is equally important for the future development of robotic construction.

## 6. Conclusions

Research on robotic on-site construction, especially for whole buildings, is still in its infancy, and future actual construction is needed to support the conclusions of simulation experiments and present many theoretical, practical, and methodological challenges. The integration of digital design and automated construction is at the core of automated building fabrication and robotic construction, which fundamentally expands the scope of traditional building construction and introduces an assembly logic for robotic automated construction to the industry [29]. While robotic on-site construction across the construction industry is still in its infancy and there are many challenges to overcome in the future, research in this highly interdisciplinary field is making progress and attempting to find solutions for robotic participation in building on-site construction [9].

**Author Contributions:** Conceptualization, L.W.; methodology, L.W. and H.F.; software, L.W. and Y.L.; validation, L.W.; formal analysis, L.W. and T.Z.; investigation, L.W.; resources, L.W. and H.F.; data curation, L.W.; writing—original draft preparation, L.W.; writing—review and editing, L.W.; visualization, L.W.; supervision, H.F. and T.Z.; project administration, H.F.; funding acquisition, H.F. All authors have read and agreed to the published version of the manuscript.

**Funding:** This research received no external funding.

**Institutional Review Board Statement:** Not applicable.

**Informed Consent Statement:** Not applicable.

**Data Availability Statement:** The data presented in this study are available on request from the authors.

**Conflicts of Interest:** The authors declare no conflict of interest.

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
