# Peer review of "Research on the Application of Mobile Robot in Timber Structure Architecture"

_sustainability, doi:10.3390/su14084681_

Round 1
Reviewer 1 Report
This paper investigated the Research on the Application of Mobile Robot in Timber Structure Architecture and it is well designed.
Introduction section must be written in a more quality way, i.e. more up-to-date references addressed. The research gap should be delivered more clearly with the directed necessity for the conducted research work. The problem statement of this study not so clear and what the different with previous work.
Please check whole manuscript grammarly.
Please add this reference to the introduction parts in terms of the biggest beneficiaries for industries that rely on manufacturing processes where parts can be moved around related studies "Touaibi, R., & Koten, H. (2021). Energy Analysis of Vapor Compression Refrigeration Cycle Using a New Generation Refrigerants with Low Global Warming Potential. Journal of Advanced Research in Fluid Mechanics and Thermal Sciences, 87(2), 106-117"
This paper can lead to further studies.
This paper can be evaluated after these revisions.
Author Response
Point 1: Introduction section must be written in a more quality way, i.e. more up-to-date references addressed. The research gap should be delivered more clearly with the directed necessity for the conducted research work. The problem statement of this study not so clear and what the different with previous work.
Response 1: The necessity of this research is demonstrated by these innovative points that distinguish it from other latest references. Until now, no robotic platform has been able to fully satisfy the requirements of an autonomous mobile robot of equal scale for the construction of complete wooden buildings (not only for structures or building components). This platform includes the integration of robotic construction technology and LiDAR positioning technology and attempts to explore the development of assembly strategies for on-site manufacturing of mobile robots for wooden buildings.
In Chapter 2, in lines 160 to 179, today, many companies are using robotic automation in on-site construction, but mostly applied to very specific subtasks. For example, the "In situ Fabricator" proposed by ETH: is a class of mobile robots specifically designed for on-site digital fabrication, but the subject of this experiment is not a complete full-size building [12]. Because of the unstructured nature of the construction environment, which makes human-robot interaction challenging, human proximity and vulnerability in interaction impose severe limitations on human and robotic activities in shared environments [27,9]. The study [28] uses multiple node robots to collaborate to meet the construction needs of large-scale structures. In our paper, we attempt to build a whole house with a limited number of robots using only one mobile arm robot. For other reasons, such as positioning. In the study [4], the research focuses on the collaboration of multiple stationary robots to complete the assembly strategy of timber trusses. Compared to stationary robots, our paper mainly emphasizes the use of mobile robots and positioning methods, which can perform a larger range of operations and are more suitable for complete building construction. In [29,30], there are quasi-fixed base full-size mobile systems capable of printing large-scale foam structures for whole-house 3D printing, but the limited movable fixed-base units do not require robotic repositioning. these challenges make automated construction using mobile robots not yet ready for the commercial market [9]. To date, no robotic platform has been able to fully satisfy the requirements for autonomous mobile robotic architectural construction of equal scale.
Point 2: Please check whole manuscript grammarly.
Response 2: Thanks to your correction, I have revised the whole manuscript for problematic grammar.
Point 3: Please add this reference to the introduction parts in terms of the biggest beneficiaries for industries that rely on manufacturing processes where parts can be moved around related studies "Touaibi, R., & Koten, H. (2021). Energy Analysis of Vapor Compression Refrigeration Cycle Using a New Generation Refrigerants with Low Global Warming Potential. Journal of Advanced Research in Fluid Mechanics and Thermal Sciences, 87(2), 106-117"
This paper can lead to further studies.
Response 3: Thanks to your suggestion, this reference has been added to the introduction parts of the manuscript at [3], in Chapter 1, in line 33.

Reviewer 2 Report
First of all, there are too much typo-mistakes in writing. In addition, the referencing was found in introduction part only. The research on application of Mobile Robot in Timber Structure architecture is not quite new, and I can't find the progress or different with previous similar works, for examples:
Samuel Leder, Ramon Webertuttgar, Dylan Wood, Oliver Bucklin, Menges, Distributed Robotic Timber Construction: Designing of in-situ timber construction system with robot-material collaboration. Conference: Association for Computer Aided Design in Architecture 2019
Kramberger A, Kunic A, Iturrate I, Sloth C, Naboni R and Schlette C (2022) Robotic Assembly of Timber Structures in a Human-Robot Collaboration Setup. Front. Robot. AI 8:768038. doi: 10.3389/frobt.2021.768038
In more detail, there are some issues need to be resolved:
- In introduction part authors mentioned "Based on this, recent research has moved in the direction of on-site digital fabrication,where buildings (or building components) are fabricated autonomously on site, generally referred to as in situ Fabrication.
Was it the novelty of the research? any similar robotic application to these scheme, please mentioned in detail and give highlighted specific purposes.
2. What is the correlation of the method utilized in research with sustainability. It should be clear from the beginning of idea.
3. In line 205:
The challenges of the experiment and future research directions are described in Chapters 5 and 6.
Which chapter??
4. Authors mentioned in line 400:
From Figure 16, it can be seen that Unit A is the least efficient and Unit E is the most efficient.
Please give justification for this statement.
5. Figure 16 and 17 have to be replotted.
6. References was found in conclusion, which is not being mentioned in the discussion part.
7. The suggestions and specific findings related with the sustainability terminology should be presented in more comprehensive in the discussion part.
Author Response
First of all, there are too much typo-mistakes in writing. In addition, the referencing was found in introduction part only. The research on application of Mobile Robot in Timber Structure architecture is not quite new, and I can't find the progress or different with previous similar works, for examples:
Samuel Leder, Ramon Webertuttgar, Dylan Wood, Oliver Bucklin, Menges, Distributed Robotic Timber Construction: Designing of in-situ timber construction system with robot-material collaboration. Conference: Association for Computer Aided Design in Architecture 2019
Kramberger A, Kunic A, Iturrate I, Sloth C, Naboni R and Schlette C (2022) Robotic Assembly of Timber Structures in a Human-Robot Collaboration Setup. Front. Robot. AI 8:768038. doi: 10.3389/frobt.2021.768038
Response: The novelty of this thesis is that, until now, no robotic platform has been able to fully satisfy the requirements of an autonomous mobile robot of equal scale for the construction of complete wooden buildings (not only for structures or building components). This platform includes the integration of robotic construction technology and LiDAR positioning technology and attempts to explore the development of assembly strategies for on-site manufacturing of mobile robots for wooden buildings.
For two example papers, in the: Distributed Robotic Timber Construction: Designing of in-situ timber construction system with robot-material collaboration, this research investigates an approach for the construction process of a modular, distributed, robotic timber construction system, this process uses multiple node robots. However, the robotic platform studied in our paper uses only one arm robot and meets its construction needs for large-scale buildings by mobile robot construction.
in the: Robotic Assembly of Timber Structures in a Human-Robot Collaboration Setup, the experimental operational system is a collaboration of multiple stationary robots that cover a workspace range of 4.6 m. The focus of the research is to explore the assembly strategy of timber and whose object of study is timber trusses. Compared to stationary robots, our paper mainly emphasizes the use and positioning methods of mobile robots, which can perform a greater range of operations and are more suitable for complete building construction.
Point 1: In introduction part authors mentioned "Based on this, recent research has moved in the direction of on-site digital fabrication, where buildings (or building components) are fabricated autonomously on site, generally referred to as in situ Fabrication.
Was it the novelty of the research? any similar robotic application to these scheme, please mentioned in detail and give highlighted specific purposes.
Response 1: In-situ manufacturing is the goal of digitally automated construction in the building industry. A building differs from a structure in that the final assembly of a building has to happen in situ——directly at its final and definite location.
In this paper, we emphasize that the experimental object is a complete building, which makes it more necessary for the building to be built on the site, rather than being manufactured in a factory like a structure or building component. To break the range limitation of stationary robot construction and based on the volume of the complete building, instead, mobile robot construction is chosen and a multidisciplinary construction platform including digital building information, mobile robot construction, and LiDAR positioning is built.
Three in situ manufacturing experiments are described in the paper Strategies for Robotic In situ Fabrication [25], including an experiment of stationary robot in situ loam aggregation, an experiment of mobile robot in situ brickwork assembly, and an experiment of mobile in situ rebar assembly. In the paper On-Site Robotics for Sustainable Construction [13], the implementation of architectural 3d printing within the scope of a large steel cable frame is explored. These experiments are exploring the challenges faced in situ manufacturing. Our paper builds on these experiments by placing more emphasis on the integration of mobile robots with positioning techniques and attempts to explore a strategy for developing assembly for the on-site manufacturing of mobile robots for wooden buildings.
Point 2: What is the correlation of the method utilized in research with sustainability. It should be clear from the beginning of idea.
Response 2: This paper opens with a description of how building automation and robotics can improve the sustainability of the industry. in lines 33 to 46: As the mindset is shifting towards sustainability, reusability, and low-carbon society, robots are becoming increasingly popular in manufacturing and industrial sectors [4]. According to Tsai and Chang (2012), permanently sustainable development is a term that is now appearing in various areas of social and economic life. Dupuisani et al. [5,6] argue that the sustainability of the industry can be improved by increasing productivity and economic efficiency. And of all sectors, the construction industry has been an important part of the world's economy and environment and is receiving increasing attention under the global sustainability development [7]. This is not only because buildings account for more than 30% of global greenhouse gas (GHG) emissions and more than 40% of global energy consumption (United Nations Environment Programme 2009) [8]. Another reason is that automation in the construction field is considered to be effective in improving construction efficiency and construction stability, reducing worker injuries, reducing waste of resources and manpower, and creating economic value.
Point 3: In line 205:The challenges of the experiment and future research directions are described in Chapters 5 and 6.
Which chapter??
Response 3: The challenges of the experiment are described in Chapter 5, in lines 490 to 505. First, the experimental simulation of the construction site environment is relatively simple, the real construction environment will be more complex and need to deal with the uncertainty of the unexpected situation. Secondly, for the positioning method using LIDAR, the errors do not accumulate because the obtained point clouds are matched concerning the reference point clouds obtained from the same initial values, however, this location method requires the surroundings to remain largely unchanged and does not exactly match the changing environment of the construction site [31]. Therefore, an attempt can be made to scan only the building components, instead of scanning the environment. However, its ability to provide the desired accuracy is debatable [25]. Third, the research object of this experiment is a wooden structure building, the robot operating system and its end-effector can complete its nailing and glue application connection method, but different building structures have different construction methods.
Based on these challenges, specific future research directions are also proposed, in Chapter 5, in lines 492 to 493: the robot can deal with the complex situation of the construction site is the focus of subsequent research. In Chapter 5, in lines 503 to 505: it is necessary to consider the adaptability between robot operation and robot end-effector under different construction methods, which means that the exploitation of end-effector is equally important for the future development of robot building. A description of the research vision for the future of robotic construction in the construction industry is in Chapter 6, in lines 507 to 509: Research in robotic on-site construction, especially for whole buildings, is still in its infancy, and future actual construction is needed to support the conclusions of simulation experiments and present many theoretical, practical, and methodological challenges.
Point 4: Authors mentioned in line 400:
From Figure 16, it can be seen that Unit A is the least efficient and Unit E is the most efficient.
Please give justification for this statement.
Response 4: In particular, this construction was done by students lacking construction process and proficiency, and the number of students working on the construction varied greatly. in this construction, the minimum number of construction personnel for Unit E is 4, and the maximum number of participants in Unit A is 13. From Figure 16, the total construction time of unit A including manual factors is the longest, and unit E is the shortest, so the efficiency of unit A is the lowest and the efficiency of unit E is the highest.
Point 5: Figure 16 and 17 have to be replotted.
Response 5: Thanks to your correction, I have replotted the two figures. Pictures are shown in the attachment
Point 6: References was found in conclusion, which is not being mentioned in the discussion part.
Response 6: References have also been added in the discussion part, with 484 line of references [5,6], 497 lines of references [31], and 499 line of references [25].
Point 7: The suggestions and specific findings related with the sustainability terminology should be presented in more comprehensive in the discussion part.
Response 7: Based on the experimental results, these specific findings on sustainability are presented in the discussion part: in Chapter 5, in lines 482 to 488, through the results of the experiment, robotic construction improves the sustainability of the construction industry in terms of increasing production efficiency and economic efficiency [5,6]. By comparing the experiment with manual construction, it can be seen firstly that the total construction time is reduced, which is a way to improve the sustainability of construction by increasing construction efficiency, and secondly that fewer construction workers are needed, and less timber is wasted, which is a way to improve the sustainability of construction by increasing economic efficiency.

Round 2
Reviewer 1 Report
accept
Author Response
Dear reviewers,
Thank you very much for your guidance.
Reviewer 2 Report
I suggest to not mention chapter, please mention the part of article properly.
Author Response
Point 1: I suggest to not mention chapter, please mention the part of article properly.
Response 1: Thanks to your correction, I have revised the chapter that mentions in line 228 of the manuscript and changed it into parts.